# CSR Unconscious Consumption by Generation Z in the COVID-19 Era—Responsible Heretics Not Paying CSR Bonus?

**Radka MacGregor Pelikánová *** and **Martin Hála**

Department of International Business, Metropolitan University Prague, Dubečská 900/10, 100 00 Prague, Czech Republic; martin.hala@mup.cz
*   Correspondence: radka.macgregor@mup.cz

**Abstract:** The COVID-19 pandemic brought a myriad of challenges and opportunities and has influenced the modern concept of sustainability as projected into the Corporate Social Responsibility (CSR) and the underlying multi-stakeholder model. The new generation of consumers, Generation Z, has progressively increased its participation in the market and its shopping trends have been impacting the entire CSR scenery. However, little is known about their attitudes, consumption preferences and expectations. In Spring 2021, this induced a pioneering case study survey involving members of Generation Z, students from a private university in Prague, focusing on their (lack of) readiness to pay any "CSR bonus". The principal research aim was to study and understand the rather surprising unwillingness of a solvent part of the new generation of consumers to support CSR during the COVID-19 era by paying at least a symbolic CSR bonus. A formal survey involving a questionnaire, replied to by 228 students, out of which 18 totally rejected the CSR bonus, was assessed via contingency tables. It was accompanied by a complementary questioning via an informal interview and glossing. This plethora of data was processed by meta-analysis and lead to an unexpected proposition: prima facie sustainability heretics denying to pay any CSR bonus can be conscious consumers and responsible and progressive supporters of the sustainability and CSR. Their rejection is a deontological cry in a desert for more transparency, trust and the rule of law.

**Keywords:** Corporate Social Responsibility (CSR); CSR bonus; COVID-19; Generation Z; sustainability

## 1. Introduction—The Roots, Evolution and Presence of Sustainability and CSR

The idea of maintaining order, enjoying appropriate consumption in a long-term basis and assuming at least the collective responsibility is intimately linked to the evolution of human society. Ancient Egypt's civilization demonstrated a commitment to it by the pyramid buildings as well as the management of the floodplain of the Nile, while during the same time in Mesopotamia it was reflected by the famous Code of Hammurabi and the management of the land between the Euphrates and Tigris. The sustainability features and aspects of this idea became obvious via Christianity. Indeed, the Bible massively covers the need for order, planning, consumption and even sustainability as such, see e.g., the story of the 7 years of bountiful harvests followed by seven years of crop failure (Old Testament) or of a set of parables about building a house or taking care of talents (New Testament). One of the reasons for the success and long-term existence of Ancient Rome was the development of such a concept of sustainability, see e.g., the laws setting, infrastructure development or channeled consumption especially during the Roman Empire period (Pelikánová 2017). The transfer of the concept of sustainability in the European Middle Ages was achieved due to the canonic unification and monastic education centers, and the renaissance of Roman law through the recovery of the *Corpus Iuris Civilis* (*Codex Justinianus*, *Digest*, *Institutes of Justinian*, *Novellea Constitutiones*) of the Byzantine Emperor Justinian I.and his famous wife Theodora in Bologna. The originally rather agricultural, state and legal dimensions have been complemented by commerce considerations. Indeed, the Hanseatic League

strongly contributed to the development of the concept of sustainability aka *Nachhaltigkeit*. The German impact was further incorporated in the 18th century via the influential book *Sylvicultura Oeconomica* by the German Colberist-Hans Carl von Carlowitz (Pelikánová et al. 2021) and by the 19th century influential book *Einfachste den höchsten Ertrag und die Nachhaltigkeit ganz sicher stellende Forstwirthschafts-Methode* by Emil André, which was initially published in Prague (Balcerzak and Pelikánová 2020). Thereafter, the concept of sustainability acquired a global dimension and became firmly linked to the *universal perpetuitas.* Namely, the move from the long-term to the eternal responsibility and from thinking regionally to globally was completed by the United Nations ("UN") in the 20th century (Schüz 2012). The 1948 UN Universal Declaration of Human Rights ("UDHR") with its human rights dimension of the social dimension of the concept of sustainability was soon confronted with rather pragmatic concerns regarding the need to balance available resources with the increasing world population (Meadows et al. 1972) and social progressive values in the context of the political awareness under the auspices of "communitarism" (Pelikánová et al. 2021). This rather settled picture was challenged by a set of crises in the 1970s and the move from Keynesian economic theory, advancing government intervention and focusing on the interaction between savings and investment, to neoliberalist economic ideology championing privatization, deregulation, austerity and reduction in government spending (Balcerzak and Pelikánová 2020). Interestingly, this evolution period of economic thinking in re the Chicago school, advancing limited government and monetarism (Milton Friedman), and even towards the modern Austrian school, advancing a mere observance of the rule of law and otherwise *laisser faire* (Friedrich Hayek), lasted until the next crises in 2007–2009. Then again the Keynesian economic theory entered into the picture. During this time of the move away from government interventionism towards the individualism and back, the UN issued a myriad of key instruments for the modern concept of sustainability, such as Agenda 21 from 1992, the Global Compact from 2000, the Millennium Declaration from 2000 with the Millennium Development Goals, etc. The two most influential of them are (i) the Report of the World Commission on Environment and Development Report: Our Common Future prepared by the Brundtland Commission, published as the UN Annex to document A/42/427 in 1987 ("Brundtland Report 1987") and (ii) the Resolution Transforming our world: the 2030 Agenda for Sustainable development ("UN Agenda 2030") (Pelikánová et al. 2021). The Brundtland report cemented the incorporation of the modern concept of sustainability based on three pillars: environmental (planet), social (people) and economic (profit) (Pelikánová et al. 2021), while UN Agenda 2030 made it potentially more actionable ((Van Tulder et al. 2016; Van Tulder and Keen 2018). Already a cursory study of the UN Agenda 2030 with its 17 Sustainable Development Goals (SDGs) and 169 associated targets implies that these policies are aimed towards the entire global society. To put it differently, UN Agenda 2030 calls for the engagement and commitment by states as well other players, including businesses—it calls for their individual responsibility regarding the impact of their endeavors on the entire society, i.e., for their Corporate Social Responsibility ("CSR"). The EU understands it very well. The EU's policies and law, such as Directive 2013/34/EU (Jindřichovská et al. 2020; Pelikánová et al. 2021) reflect the pragmatic fact that the modern concept of sustainability is ephemeral and illusory, perhaps even futile, without the employment of a multi-stakeholder model and cross-sector partnership (Van Tulder et al. 2016; Van Tulder and Keen 2018) leading to the CSR endorsed by businesses and supported by other stakeholders, including consumers. However, the reality is more complex and the current COVID-19 pandemic and its multi-spectral impact along with the emergence of the new consumer generation—Generation Z makes it even more Byzantine.

It must be emphasized that, unlike the global modern concept of sustainability with continental roots (Roman law) currently backed by the international law (UN), regional law (EU) and state laws, CSR has a very different nature and history. In contrast to sustainability, CSR basically did not exist until the 19th century and it emerged in a rather subdued manner in the US in the context of the first antitrust wave (the Sherman Antitrust Act of 1890),

the criticism of deplorable working conditions and the reinforcement of philanthropic moves. During the Great Depression in the US, the increased sustainability concern was matched with the emergence of Codes of Ethics (McDonald 2009) as self-imposed business regulations under the command of the ephemeral and subjective philosophical notion of rightness by their subjects (Babri et al. 2021). Since each Code of Ethics is a product of one business and applies only to that business, it does not rely on a multi-stakeholder model (De Bakker et al. 2019). Their violation does not engage a legal liability, but may lead to certain sanctions and other negative consequences, such as a lack of personal appraisal or negative publicity (Kaptein 2011; Oladinrin and Ho 2016). Basically, both CSR and Codes of Ethics are answers to an ethical call which is a subject of various appreciations, and do not lead to legal liability (Balcerzak and Pelikánová 2020). Crises and scandals are often milestones of their parallel evolution, see e.g., various catastrophic business and investment events in 2002 (Cerchia and Piccolo 2019) or the financial crisis 2007–2009 or the current COVID-19 pandemic (Balcerzak and Pelikánová 2020). It cannot be overstated that a crisis is both a challenge to the existing framework and also an opportunity to change, i.e., businesses facing a crisis are impelled to (re)consider and re(state) their identity, priorities and self-presentation (Kovoor-Misra 2009). For example, the current COVID-19 pandemic is a true test of the foundation of basically each and every business as captured by Codes of Ethics. This test shows whether a business is set up to align with the idea of Albert Einstein about the effect of a crisis to stimulate human progress, inventiveness and innovations (D'Adamo and Lupi 2021), see recent digitalization and pharmaceutical achievements.

Nevertheless, two particular milestones in the evolution of CSR need to be emphasized. Firstly, the publication of the landmark book Social Responsibilities of the Businessman by Howard R. Bowen in 1953, which spelled out that the largest US businesses are centers of power and decision-making, touch the lives of all and should be at least to some extent responsible for that (Carroll 2016). Secondly, the four-parts definition of CSR as an economic, legal, ethical and discretionary (philanthropic) responsibility was stated in 1979 and depicted via the famous Carroll's pyramid in 1991 (Carroll 2016). However, as indicated above, this does not mean that sustainability is projected into the CSR of all European businesses and enjoys the unilateral support of all stakeholders, including Generation Z consumers. Indeed, both the entire CSR is challenged by partisans of the traditional theory and opponents of the stakeholder approach and individual parts of the CSR pyramid are undergoing a critical scrutiny (Pelikánová et al. 2021). The ultimate judges, customers, react in a very sensitive manner and their responsivity to CSR is definitely modified by the COVID-19 pandemic. The flagship of their attitude is carried on by the ambitious new coming cohort of consumers which have both the resources and willingness to influence the destiny and operation of CSR—Generation Z (Pelikánová and MacGregor 2020a). Indeed, for the present and future, it is critically important for states and their laws (Pelikánová and MacGregor 2020b) as well as for businesses and their strategies to understand the willingness and readiness of members of Generation Z to support sustainability and CSR. Ethical business behavior is positive for investors, satisfying suppliers and customers as well motivates employees (Bočková et al. 2012). Perhaps the most important is to detect the parameters and circumstances when a total failure occurs. Namely, as always, crises magnify issues and their impacts and the COVID-19 setting leads to a total rejection of CSR by a section of the solvent and educated members of Generation Z. This is extremely worrisome and the plain condemnation of these members while labeling them greedy and irresponsible is a fatal short-circuit. In contrast, it is extremely illuminating to empirically explore who are these "CSR heretics" and to engage in a deeper understanding why they reduce themselves to such a "barbarian" behavior. Hence, the principal research aim is to discover and assess the surprising, if not shocking, unwillingness of a solvent part of the new generation of consumers, Generation Z, to support CSR during the COVID-19 era by paying at least a symbolic CSR bonus. Such an aim is to be addressed via a two-stage case study: (i) a formal questionnaire survey processed via contingency tables and (ii) informal complementary interviews which are a part of a longitudinal project focusing on both

qualitative and quantitative aspects of the generation Z CSR readiness. Consequently, after this introduction (I), the pertinent background is to be explored (II) and the case study is to be performed by entailing survey and contingency tables (III). Such data and methods lead to interesting results identifying these "CSR heretics" from Generation Z. Under the auspices of the discussion, a pioneering proposition emerges and completely turns the lens—shopping with a conscience can be done without paying a CSR bonus (IV). In sum, the massively advanced pro-CSR and pro-CSR bonus attitude under the auspices of conscious consumption is not per se the only way and it can even become an ineffective and/or inefficient and/or illegitimate venue to sustainability (V). Responsible CSR heretics are at the gates with an important message.

## 2. Theoretical Background—The Carroll's CSR Pyramid Temple, Believers and Heretics before and during the COVID-19 Era

CSR is a private reflection of the public concept of sustainability, i.e., it is a systemic reaction of each individual business to the global and general call for a balanced management and use of resources (Pelikánová et al. 2021). It represents a dialogue and interaction between businesses, corporations and their stakeholders (Małecka et al. 2017), including customers (Pelikánová et al. 2021) within the given legal framework and set state policies (Šebestová et al. 2018). Since, within the EU, the CSR leads to only a rather weak reporting duty of certain strategic entities (Pelikánová et al. 2021), it is generally in the discretion of European businesses, including Czech businesses, whether and, if yes, how, where and to what extent they will do it (Pelikánová et al. 2021). Their strategic choice about it is determined by a myriad of internal and external factors. To such internal factors belong the mission and key values of the business as well as preferences of its very internal stakeholders. To such external factors belong the legislative and policy framework as well as preferences of external stakeholders, such as consumers (Pelikánová and MacGregor 2021). Since typically these factors and underlying preferences partially contradict each other and need to be balanced, see e.g., R&D asking for high investment (Pelikánová et al. 2021), at the very end, the turning point is a business-philosophic attitude of the given business to follow either traditional theory or stakeholder theory. To put it differently, the strategic choice of each and every business whether to engage in CSR, and, if yes, to what extent, is strongly influenced by the adherence of the business management to the traditional theory or stakeholder theory (Pelikánová et al. 2021).

Pursuant to the traditional theory, each business has only two responsibilities—(i) to respect the law while behaving in good faith and (ii) to seek material gain, see publications by Theodore Levitt about the profit maximization command (Levitt 1958) or by Milton Friedman about the irrelevancy of social issues for business conduct (Friedman 1962). This traditional attitude along with real life observations makes certain businesses in the 21st century perceive a commitment to sustainability via CSR as a negative burden generating costs without returns, i.e., a waste preventing profit maximization (Friedman 2007). For them, CSR is a Pharisaic endeavor pretending to overcome the agency problem without an impact on financial performance (Strouhal et al. 2015). There are even voices suggesting that it is orchestrated by forces making from businesspeople "unwitting puppets", "undermining a free society these past few decades" and bursting in "unadulterated socialism" (Friedman 2007). Despite this rather strong language, the CSR sceptics are not delirious and selfish cormorants/gannets/graspers, because a number of empirical studies revealed the negative impacts of CSR activities and spending by indicating that CSR practices can become a powerful tool in the hand of the top management (MacGregor et al. 2020), generate unnecessary costs, cripple financial results (Barnett 2007) and undermine the competitive advantage (Scherer and Palazzo 2011). This leads to a marketing, which undermines the strength of the information about inventiveness for stakeholders (Roszkowska-Menkes 2017) and about the creation and bringing of values to stakeholders (Zollo et al. 2018). In contrast, such type of marketing is founded upon the conviction about the almost inevitable communication noise, inducement to information ignorance and resultant confusion of stakeholders, in particular consumers (Chauhan and Sagar 2021). With a touch of exaggera-

tion, it can be suggested that there is an overlap between, on one hand, these traditionalists and CSR sceptics and, on the hand, followers of the Chicago school and modern Austrian school, i.e., anti-Keynesians. Similar to the changes of high and low times for schools of economics, after the time of the glory of the traditional profit maximizers CSR sceptics, there came the era of the stakeholder attitude and CSR promotion.

Pursuant to the stakeholder theory, the business engagement with CSR leads to a value creation and an increase in performance, profit and market share (Ting et al. 2019). This is further magnified by the influence of business ethics and their increasing role (Sroka and Lörinczy 2015). Allegedly, the endorsement of shared value policies and principles linked to CSR should lead to "a more sophisticated form of capitalism" (Porter and Kramer 2011), while the evolution should go from the CSR cultural reluctance over to the CSR cultural grasp to a CSR cultural embedment (Olšanová et al. 2018). In particular, CSR is arguably a terrific opportunity, a win-win vehicle, an impulse for improvement in all three spheres of sustainability (economic, environmental and social), an instrument to improve, tangibly, one's own financial performance (El Ghoul et al. 2011) as well as an intangible, the business reputation (Gallardo-Vázquez et al. 2019) and branding (Osei-Tutu 2019) and a foundation for the marketing (Adamska 2020) and other strategies of each and every business (Pelikánová et al. 2021). Indeed, the positive impact of CSR extend not only to the business performance and its results, but as well bring social advantages such as employee stability, AKA human resources retention, improvement of local community relationships and even the attraction of social and ethical investors (Goss and Roberts 2011) and customers (Bhattacharya and Sen 2004). This should support a set of values—from practical inventiveness (Al-Jundi et al. 2019) to more ephemeral moral principles (Balcerzak and Pelikánová 2020). A proper management should carefully take advantage of that and be clear and concise in order to avoid confusing consumers, match with their expectations (Al-Jundi et al. 2019) and enhance their awareness (Chauhan and Sagar 2021), and ideally to enhance their readiness for a mutual commitment (MacGregor et al. 2020). Ideally, the average customer will be impressed by the CSR of a business and will gladly pay something extra, i.e., a (circular) premium often called the CSR bonus and recent pre-COVID empirical studies confirm the customers'readiness to pay extra for pro-sustainability and/or pro-CSR goods or services (D'Adamo and Lupi 2021). Arguably, the most responsive in this respect are working consumers (Zhang et al. 2021) and young consumers. Several studies with empirical analyses confirm that and propose that CSR brings benefits for all stakeholders and enhances financial performance (Rowley and Berman 2000; Turečková and Nevima 2020). However, their results are not conclusive, because other studies bring totally different results, i.e., there is an ongoing discourse about the positive or negative or neutral impact of CSR on business performance, especially financial performance (McWilliams and Siegel 2000).

Despite the inconclusiveness of performed studies and the ongoing dispute between traditional theory supporters and stakeholder theory supporters, there is no doubt that businesses in the 3rd decade of the 21st century must properly reconcile profitability, growth and social relationships (Rodriguez-Fernandez 2016) and must deal with CSR (Zaušková and Rezníčková 2020), especially during challenging times of crises (Derevianko 2019). Indeed, crises should stimulate progress and innovation changes (Schumpeter 1934) and these changes should be welcomed by all stakeholders, especially by customers as principal targets of the creation by all businesses (Drucker 1973). To put it differently, marketing has to produce results, otherwise it is a mere cost (Drucker 1973), while proper marketing linked to innovations and "right" values (Al-Jundi et al. 2019) is a great fundament of every business management (Drucker 2015). Naturally, the entire point of marketing is to produce effective, efficient and persuasive information, disseminated to its ultimate addresses.

Consequently, during the COVID-19 pandemic and immediately after it, the question is not about the quantity (presence), but about the quality (content) of the well communicated CSR as the reply of businesses to the increased of the social sensitiveness to about ethical (Sroka and Szántó 2018), social (Mallin 2018) and environmental (Krause 2015) issues (Pakšiová 2017). Therefore, effective, efficient, legitimate and properly oriented and

reported CSR is an expense which could (and should) be compensated for by a myriad of positive effects, such as advertising and brand image improvement (Han et al. 2010), fixation of stable revenues from loyal clients, increase of employee productivity (Ikram et al. 2019), risk reduction (Sharfman and Fernando 2008) and diminishment of capital costs (Galbreath 2013; Pakšiová and Oriskóová 2020). Therefore, a good CSR has five features, i.e., these five are conditio sine qua non for a proper CSR: (i) being good i.e., effective, (ii) well materialized (efficient), (iii) conform to state and stakeholders' preferences (legitimacy), (iv) moving towards targets (oriented) and (v) disseminated (reported). Such a CSR is cost effective (Osei-Tutu 2019) and leads to black numbers (Pelikánová et al. 2021). It is argued that CSR which does not go for correct values (ineffective) or is wrongly materialized (inefficient) or is detached from ethical and law expectations (illegitimate) or does not hit its target audience (wrongly oriented) or the information about it is not disseminated (ill reported) is a plain waste, sometimes even counter-productive and repulsive (Pelikánová et al. 2021). However, this argument is so far minoritarian, because the leading voices worship the CSR temple in the shape of the famous Carroll's pyramid (Carroll 2016), while assuming that businesses will set a CSR bonus, customers pay it and this will be an incentive tool for even more engagement with CSR and for financing Carroll's pyramid. Academic proponents of such a vision have predominantly a background in management, including business ethics, in environmental studies and in political science, and only seldom in anthropology, history and law and only exceptionally are practitioners (De Bakker et al. 2019). This explains a lot.

Certainly, Carroll's pyramid brings a radical answer to the Bowen question "what responsibilities to society may businessmen reasonably be expected to assume?" by offering four vertically positioned and interrelated responsibilities: economic, legal, ethical and philanthropical. This set of four responsibilities addresses expectations by stakeholders, especially customers, and creates a foundation for the particular business (Carroll 2016). At the same time, via the multi-stakeholder model with multi-stakeholder initiatives on sustainability (De Bakker et al. 2019), the customers, especially members of Generation Z, should appreciate that and support it by paying a CSR bonus. However, Carroll's pyramid is neither a static structure with four strictly separated vertical layers nor a universal instrument taking care of all sustainability issues. Instead, Carroll's pyramid is based upon the humble recognition that there is an overlap and interaction of the economic responsibility towards shareholders and employees, a legal responsibility towards employees and customers to go for the letter of law (positivistic codified ethics), an ethical responsibility towards customers to go for the spirit of the law and a philanthropic responsibility to the community to "give back" to the society (Carroll 2016). The mantra is crystal clear—these responsibilities overlap (see e.g., how ethics permeates the pyramid) and there are tensions and trade-offs between them (Carroll 2016). Therefore, Carroll's pyramid is just a starting point to balance what is required, expected and desired by society and the success of such balancing means a good CSR, while a disbalance means a bad CSR. Hence, it is not so much about the quantity/existence, but the quality of CSR setting and application—and the ultimate judge is the ephemeral average customer.

Certainly, Vox populi, vox Dei, but the voice of the people, or more specifically of customers, is not a unanimous and fixed tenor. Indeed, customer groups differ in their attitudes and appreciation regarding CSR (MacGregor et al. 2020) and even each group changes its attitudes and appreciation over time, especially if a crisis hits (Pelikánová et al. 2021).

In this respect, it is highly relevant to perform a longitudinal observance of a solvent customer group with a growing impact—Generation Z which is socially aware, e-literate (Turner 2015) and Internet and social media dependent (Bassiouni and Hackley 2014; Choi et al. 2021). Generation Z organically continues trends launched by the preceding Generation of Millennials which established the four components cyclic customer journey: connect, explore, buy and use in the context of the real and electronic universe (Mele et al. 2021). For both Millennials and Generation Z, online and offline interactions blur the boundaries

between the physical and digital world (i.e., phygital) (Mele et al. 2021). They both are well-known for their drive for authenticity and its importance for consumption choices (Nunes et al. 2021). However, it should not be overlooked that Millennials had a very structured childhood managed by their Baby Boomer parents tending to practice "helicopter" parenting, while Generation Z is a product of the post-9/11 world, a time of economic lability, political polarization, and multiple foreign wars (Talmon 2019). These circumstances led to Generation Z becoming very pragmatic (Talmon 2019) and this pragmatism becomes projected in their consumer behavior (Pelikánová and MacGregor 2020a).

An interesting section of Generation Z is composed of ambitious young people wanting to achieve a good tertiary education (and ready to pay for it) in order to get well-paid jobs in the future (Dvouletý 2017). At the same time, their determination, social and digital awareness might induce the idea that they will automatically opt for CSR and, if not totally selfish, then they should be ready to pay a CSR bonus (D'Adamo and Lupi 2021; Pelikánová et al. 2021). However, we should keep in mind that Generation Z is very much oriented towards digital media, spends a great deal of time looking for information posted on various portals or in social media (Choi et al. 2021) and considers propositions by various influencers (Gajanova et al. 2020). They definitely do not reduce themselves to conventional content analysis of reports (Vourvachis and Woodward 2015).

Undoubtedly, a proper CSR leads to an increase of legitimacy and facilitates customer engagement (Carroll 2016), but members of Generation Z are not only more inclined to actively support CSR than older consumers (D'Adamo and Lupi 2021), but as well they are very autonomous in deciding what they perceive as a good or bad CSR and what and how they will support it. They do not hesitate to look for information via physical as well as digital platforms (Mele et al. 2021) and gladly share their opinions, experiences and attitudes via various social and other media (Choi et al. 2021; Pelikánová and MacGregor 2020a). Members of Generation Z know very well that businesses can do something completely different from what they state, see the impression management issue (Meyer and Rowan 1977; Roman et al. 2019). They carefully check if the business' daily decisions as well as longer term strategies are derived from the preferences of the population and are a part of a chain of accountability, i.e., whether there is a legitimacy (De Bakker et al. 2019) or not. If not, then such a business engagement is not a CSR deserving reward by Generation Z. Unsurprisingly, Generation Z is not easily influenced by simplistic marketing tools, such as a selection of the color of the labels (Samaraweera et al. 2021). Therefore, we need to be extremely cautious before making conclusive statements about what Generation Z wantst and is ready to pay for, and how much. They and their opinions need to be included in the CSR, otherwise they will perceive a problem of a lack of legitimacy (De Bakker et al. 2019).

Indeed, propositions and conclusions about customers' behavior, especially behavior of Generation Z, are even more speculative during turbulent times. Since crises magnify differences and launched trends, the COVID-19 pandemic should be a great test case for the consciousness of consumption by Generation Z. COVID-19 aka SARS Covid 2 (Manojkrishnan and Aravind 2020) emerged in its original coronavirus form around 2002 (Rasool and Fielding 2010) and its version was called MERS in 2012 (Manojkrishnan and Aravind 2020). In 2020, the COVID-19 virus was declared a global pandemic by the World Health Organization (Armani et al. 2020). This unprecedented pandemic hit each and every nation and economy world-wide and has brought a global economic downturn which has not been experienced since the 1870s and the worst economic crisis since the 1930s (Pelikánová et al. 2021) along with a dramatic disruption of the capital market (Pardal et al. 2020) as well as the stock market. The entire society has suffered a set of negative impacts and pre-existing inequalities have expanded (Ashford et al. 2020) threatening the sustainability of the entire society and in particular the protection of the environment and the well-being of customers confronted by both the shortage of certain goods and services and the increase of their prices. Sadly, the large majority of businesses and businesspeople had no prior proper planning for that and were exposed to significant risks. Namely, only 20% of executives were confident that their companies were prepared to respond to

such a risk, while ultimately 94% of the Fortune 1000 companies have suffered serious consequences.

All this induced the EU, represented by the European Commission president, Ursula Von der Leyen, to make a set of crucial statements, while emphasizing that "We must not hold on to yesterday's economy as we rebuild" (World Economic Forum 2020) and "We have to push for investment and reform—and we have to strengthen our economies by focusing on our common priorities, like the European Green Deal, digitalization and resilience" (European Commission 2020). Indeed, the EU is attempting to reconcile competition concerns (Pelikánová et al. 2021), the drive for technological and other potentialities (Kotlebova et al. 2020) and the demands of sustainability via CSR (Landmesser 2019) in the COVID-19 context—the pandemic is not only a threat, but as well an opportunity for CSR (D'Adamo and Lupi 2021; Pelikánová et al. 2021). Allegedly, Albert Einstein stated: " . . . it is crisis that brings progress. It is in crisis that inventiveness, discovery and great strategy are born". (D'Adamo and Lupi 2021).

The EU uses rather softer instruments for that, i.e., policies rather than strict mandatory law provisions (Pelikánová and MacGregor 2021). Consequently, businesses have a rather broad freedom to select their strategies and manner how they will report about it (Pelikánová and MacGregor 2020b). Naturally, their decision-makers as proper agents should make the best decisions of such businesses and during a crisis, such as the COVID-19 pandemic, they need to be extremely careful with respect to the attribution theory and its bridging to the stakeholder model. Since attributions are the result of the fundamental cognitive processes by which people ascertain cause and effect so that they can solve problems and become more efficacious in their interactions with their environments (Martinko et al. 2007), the behavior of businesses needs to balance and reconcile the short-time crisis survival command and the long-term sustainability demand. Naturally, the attribution is intimately linked to the motivation and the prevailing academic stream is convinced about the market-based approach, arguing that market actors themselves, especially consumer-oriented businesses, are interested in the adoption of CSR to increase their reputation and both consumer and investors interest (De Bakker et al. 2019; Borseková et al. 2021). Only a few scholars challenge the assumption that businesses go for CSR due to their market-based motivation, i.e., hoping to achieve long-term financial returns, and so academia is convinced that utilitarianism dominates (De Bakker et al. 2019). However, at least two alternatives should be considered— (i) deontological reasoning aka Kantian demand for a duty and for judging action not based on the consequences and outcome but based on its motivation and (ii) an integrative social contracts theory based on the Rousseau mutual and reciprocal freedom observation in our global and diverse universe (De Bakker et al. 2019).

As already mentioned, the obvious judge of the correctness of the selected road (effectiveness), the journey on it (efficiency) and the conformity of such a road and journey with logic, justification and law (legitimacy) is the customer. He or she will explore not only economic, but as well legal and ethical dimensions and accordingly will decide whether he will purchase goods and/or services from such a business and if yes, then for how much. If such a customer is a member of Generation Z, then it must be emphasized that he or she will look for information and the media (Choi et al. 2021), which is typically excluded from deliberative analyses, will play an important epistemic role (De Bakker et al. 2019). Indeed, preferences by the generation of consumers able to truly speak with their purchasing decisions in the next decades, i.e., members of Generation Z with a sufficient financial potential (Pelikánová and MacGregor 2020a), are indispensable and critical for the multi-stakeholder model supporting sustainability and CSR. They are not passive bystanders, instead they are ready to become active participants ready to take advantage of multi-stakeholder initiatives on sustainability (De Bakker et al. 2019). Hence such consumers paying such a "CSR bonus" for products or services of such businesses can significantly help to make CSR pay off, not only environmentally and socially, but economically (financially) as well. At the same time, Generation Z is not a homogenous group and obviously includes various subgroups of members with similar behavior patterns. Considering the conventional categorization

and variable setting, gender, age and origin are typical distribution instruments splitting a heterogenous student group into a set of more homogenous groups (Bećirović et al. 2018). In particular, gender-based differences and resulting strategies selections have attracted massive interest (Chavez 2001; Mayer 1996). The distinction and analyses based on age and nationality are as well commonly acceptable and often used (Primack et al. 2009; Marquardt and Herrera 2015).

Regardless of their categorization and variables selection, by financially rewarding businesses for their CSR, consumers can assist in turning CSR into a competitive advantage (Pelikánová and MacGregor 2020a). The question is, do they do that in real life? In particular, are financially sufficiently strong members of Generation Z inclined to pay a CSR bonus during the COVID-19 era? Do they engage in conscious consumption or are they greedy and selfish? Perhaps the CSR attitude and support via a CSR bonus is not binary and in addition to support-altruist and not to support-egoist there is a third option. So who are these members of Generation Z rejecting to pay a CSR bonus during the solidarity era of the COVID-19 and why they are reducing themselves to be heretics? Or are they conscious consumers in the COVID-19 era without paying the CSR bonus?

## 3. Materials and Methods—A Case Study Survey with Contingency Tables

The materials, data and methods were selected to address the principal research aim—to discover and assess the surprising unwillingness of a solvent part of the new generation of consumers, Generation Z, to support CSR during the COVID-19 era by paying at least a symbolic CSR bonus. The case study format was selected because it reflects the longitudinal as well as investigation demands (Yin 2008). The CSR studies generally focus on qualitative data (De Bakker et al. 2019), but this case study entails as well conceptual and quantitative research. Due to the focus on the consumers perspective, the content analysis of reports will be pushed back (Vourvachis and Woodward 2015).

Namely, a pioneering case study was employed to survey students from a private university in Prague, in the Spring of 2021, focusing on members of Generation Z declining to pay anything extra in the price as a reward for the CSR of the particular business. Consequently, the fresh primary data were collected by asking 300 students aged 19–28 years, i.e., basically meeting the definition of Generation Z. They all attend Business and Law courses at a private university in Prague, do not benefit from any scholarship and can be considered currently (or at least potentially) solvent. Therefore, the sample was sufficiently homogenous and matching the criteria of the new wave of consumers, and perhaps even some future managers, in Central and Eastern Europe. Hence two steps of the process of collecting primary data (formal survey and informal interview) could be performed while maintaining academic robustness.

Firstly, these students taking this paid tertiary education were invited to complete a formal survey. They were asked to enter their age, gender, nationality and to indicate their readiness to pay a "CSR bonus", in a percentage format, for a product or service with a strong CSR background as opposed to a product or service with a neutral CSR provenience. The exact wording of the survey text was as follows: "Considering the current situation and global society challenges, would you please indicate how much extra in % you are open to pay for an identical product/service of a business which goes strongly for sustainability and CSR as opposed to a neutral business, i.e., doing nothing for or against sustainability and CSR (nothing for or against the environment, neutrally treating employees, neither helping nor damaging society, etc.). Hence, there is no difference between the compared outcomes, they have the same quality and they differ only regarding their sources and you might be open to pay e.g., 20% extra in the case of a business going strongly for CSR . . . or maybe you do not want to pay anything extra. It does not matter, just please indicate below your honest opinion". The survey was announced during the online lecture and the above indicated text was posted via MST and SIS posting while an online oral presentation of the survey was performed and any questions raised by the students-respondents were answered. The purchase of a pair of white tennis shoes was given as an example, i.e.,

students were told that they need to buy a pair of white tennis shoes and on the shop shelf there are three similar pairs with the same quality—the first one is produced by a CSR neutral business, the second one by a business moderately going for CSR and the third one by a business strongly committed to CSR.

In total, 228 students matched the Generation Z criterion and at the same time provided a full reply (identification—age, gender, nationality; % CSR bonus). For the purpose of the analysis, we had to segregate the students into suitable category groups. The segregation based on age was done as follows: 19–20 years (42 respondents), 21–22 years (63 respondents), 23–25 years (84 respondents) and the remaining 26+ years (39 respondents). The segregation based on gender was done obviously for males (110 respondents) and females (118 respondents). The segregation based on origin entailing over 30 states was the most challenging and ultimately it was done by splitting countries based on the GDP and regional determination into six category groups: Czech Republic (92); Developed countries (13); Other countries (13); Post-Soviet countries (47); Post-communist countries (17) and Russian Federation (46). All involved students had to pay for their business and law study, i.e., there were no scholarship recipients. The processing of primary data gained from such a formal survey employing an investigative questionnaire led to the binary format of answers (to pay any CSR bonus—yes vs. no) which implies the option of the analysis via logistic regression. Logistic regression is a specific type of analysis employing logistic models and addressing the probability of a certain class or event while taking advantage of the binary dichotomy, i.e., assigning values 0–1 (Peng et al. 2002). Indeed, logistic regression is based upon statistical models which use a logistic function to model a binary dependent variable along with independent variables (predictors) which are either categorical, or continuous) (Tolles and Meurer 2016). However, the pertinent formal survey revealed (only) 18 students rejecting the CSR bonus completely, i.e., less than 8% of all respondents (228). This data could not be in a statistically important manner assessed as influenced by the age and/or gender and/or origin of respondents. Boldly—18 respondents and less than 8% are way too low numbers for academically robust logistic regression models. Nevertheless, still the provided data can be at least indicative, and this especially if contingency tables with the triad age-gender-origin are used and null hypotheses are considered (H0—age/gender/origin has no impact on the readiness to pay a CSR bonus). The possible fragmentation and size insufficiency can be at least partially offset by the employment of an informal anonymous interview questioning and processing via a holistic meta-analysis (Glass 1976; Schmidt and Hunter 2014).

Secondly, anonymous informal interviews about the reasons and motivation were done with students rejecting the CSR bonus. Namely, 210 of these 228 students indicated that they would pay such a CSR bonus, typically between 10% and 30%. However, particular attention was merited by the 18 students totally rejecting the CSR bonus, i.e., indicating 0% as their CSR bonus. This clearly called for a thorough methodologic processing, especially since these 18 denials matched with trends regarding a similar pool of respondents which were already observed and published immediately before the COVID-19 pandemic (Pelikánová and MacGregor 2020a). Therefore, these 18 were approached and engaged in an investigative discussion exploring why they reject a CSR bonus. The processing of the collected feedback was done in a Delphi method style (Okoli and Pawlowski 2004), i.e., a panel of three experts in this field (RKM, EDC and LM) was formed and each member of the panel interviewed each of these 18 students while following provided guidelines and then they prepared a scoring and glossing chart. Then all three experts conferred and compared their 1st round charts for each student and moved to the 2nd round of interviews and a conference to avoid any significant differences and discrepancies. The fact that each student was interviewed twice by each expert and the experts proceeded in a unified manner and conferred in order to reconcile the results reduced, but naturally did not avoid, inherent subjectivity.

Finally, it is to be underscored that this "COVID-19 era" (i) formal questionnaire survey processed via contingency tables and (ii) informal complementary interviews were

a part of a longitudinal project focusing on both qualitative and quantitative aspects of the generation Z CSR readiness. Since the meta-analysis is a quasi-statistical analysis of a broad pool of data from individual studies with the goal to reconcile them and to integrate their findings (Silverman 2013), it allowed for the longitudinal as well as multi-jurisdictional comparison. This new study from the COVID-19 pandemic blended neatly with prior studies, allows a deeper understanding via the employed glossing and Socratic questioning (Areeda 1996) and offers new perspectives about the CSR determinants of generation Z during our current challenging era. Prima facie sustainability heretics denying to pay any CSR bonus and being from other than classic developed countries can still be conscious consumers and very responsible and progressive supporters of the sustainability and CSR.

## 4. Results—Formal Survey and Informal Interviews

The assessment of a questionnaire, replied to by 228 students in Spring 2021, revealed that 18 of the students totally rejected the CSR bonus. Namely, 7.9% of students indicated 0% for the CSR bonus. This led to a logical question whether this denial of the CSR bonus payment depended upon the age, gender or the origin, or was at least influenced by them in a statistically significant manner. Due to the low number of these students, the processing and modeling via logistic regression were neither convincing nor satisfactory. However, contingency tables brought interesting insights meeting expectations regarding solid results, and this especially if complementary questioning and glossing based on the meta-analysis was performed.

Firstly, the null hypothesis about age was analyzed based on the contingency table, see Table 1—$H_0$ age. It revealed that the inclination to pay or not to pay the CSR bonus statistically depended upon the age, i.e., the relative frequency of students declining to pay any CSR bonus differed in age groups. The $H_0$ age could be clearly rejected, because the *p*-value is very low (0.049). Further, Table 1 shows that the biggest difference was between the youngest group (83.3% respondents for the payment of CSR bonus) and the oldest group, which was at (or even beyond) the edge between Generation Z and millennials (100% respondents 26+ for the payment of CSR bonus).

**Table 1.** Contingency table $H_0$ age—the frequency of no CSR bonus payers is NOT the same in age groups (age matters).

| Age Group | CSR Bonus = 0 | CSR Bonus > 0 |
|---|---|---|
| 19–20 | 7 (16.7%) | 35 (83.3%) |
| 21–22 | 5 (7.9%) | 58 (92.1%) |
| 23–25 | 6 (7.1%) | 78 (92.9%) |
| Total | 18 (7.9%) | 210 (92.1%) |
| $X^2$ test | $X^2$ = 7.85 and N = 228 | df = 3 and *p* = 0.049 |

Source: Prepared by authors based on the survey.

Secondly, the null hypothesis about gender was analyzed based on the contingency table, see Table 2—$H_0$ gender. It revealed that the inclination to pay or not to pay the CSR bonus did not statistically depends upon the gender. Namely, 8.5% of females and 7.3% of males from the given pool of student respondents did not want to pay anything extra, i.e., their CSR bonus was 0. The $H_0$ gender could not be rejected, because the *p*-value is very high (0.737).

**Table 2.** Contingency table $H_0$ gender—the frequency of non-CSR bonus payers is the same by males and females (gender does not matter).

| Gender | CSR Bonus = 0 | CSR Bonus > 0 |
|---|---|---|
| Female | 10 (8.5%) | 108 (91.5%) |
| Male | 8 (7.3%) | 102 (92.7%) |
| Total | 18 (7.9%) | 210 (92.1%) |
| $X^2$ test | $X^2 = 0.113$ and $N = 228$ | $df = 1$ and $p = 0.737$ |

Source: Prepared by authors based on the survey.

Thirdly, the null hypothesis about origin was analyzed based on the contingency table, see Table 3—$H_0$ origin. It revealed that the inclination to pay or not to pay the CSR bonus did not statistically significantly depends upon the origin. Namely, the H0 origin could not be rejected, because the *p*-value was very high (0.697). However, in contrast to the non-rejection of $H_0$ gender, at least indices of differences could be detected. Indeed, Table 3 shows that 100% of students from developed countries were ready to pay a CSR bonus while only 87.2% from post USSR countries were ready to do so.

**Table 3.** Contingency table $H_0$ origin—the frequency of no CSR bonus payers is similar (but not the same) for students of different origin.

| Origin | CSR Bonus = 0 | CSR Bonus > 0 |
|---|---|---|
| Czech Republic | 6 (6.5%) | 86 (93.5%) |
| Developed countries | 0 (0%) | 13 (100%) |
| Other countries | 1 (7.7%) | 12 (92.3%) |
| Post-Soviet countries | 6 (12.8%) | 41 (87.2%) |
| Post-Communist countries | 1 (5.9%) | 16 (94.1%) |
| Russian Federation | 4 (8.7%) | 42 (91.3%) |
| Total | 18 (7.9%) | 210 (92.1%) |
| $X^2$ test | $X^2 = 3.02$ and $N = 228$ | $df = 5$ and $p = 0697$ |

Source: Prepared by authors based on the survey.

These results were not conclusive and rather indicative, i.e., they pointed to the significance of age, insignificance of gender and semi-significance of origin. However, they were not providing deeper information about respondents rejecting a CSR bonus and about their motivation. This insufficiency could be at least partially offset by informal interviews and additional questioning, i.e., by holistically asking these respondents why they did not want to pay any CSR bonus. The veracity of their answers was improved by the anonymity. Anonymous feedback from these 18 respondents included the following statements:

- COVID-19—"We are in the pandemic/in the crisis/in the war and CSR is not the priority";
- information asymmetry—"I do not have (access to) (sufficient) information about the CSR of particular businesses";
- trust deficit—"They all lie about their CSR";
- denial of engagement—"This is not my task (business) to check their CSR, state (authorities) should do so";
- lack of interest—"I have more important stuff to do than check CSR of businesses";
- lack of resources—"I have no extra money to pay for that";

Certainly, feedback provided by 18 respondents was far from being statistically reliable, but still it was worthy of exploration. Indeed, it brought out a rather unexpected proposition that these "heretics" rejected CSR bonus, but not sustainability and CSR as such. Indeed, they were responsible consciousness consumers, but not in the institutionalized conventional manner.

## 5. Discussion

The performed case study leads to a very general proposition that the new generation of financially strong and solvent consumers, i.e., members of Generation Z which are studying and paying for Business and Law courses at a private university in Prague, is open to pay a CSR bonus as a demonstration of their commitment and willingness to support the CSR of businesses via a multi-stakeholder model and multi-stakeholder initiatives on sustainability. Indeed, 92% of such respondents are open to pay at least a small CSR bonus. At the same time, it must be admitted that the size and composition of the survey pool and the statistical modelling were not robust and that the entire case study has a potential to offer rather indicative than conclusive propositions. Nevertheless, this proposition appears to be sufficiently relevant, i.e., it is not pioneering and basically matches with the prevailing tenor based on the stakeholder theory (Małecka et al. 2017; Sroka and Lörinczy 2015; Ting et al. 2019), the importance of the authenticity concept (Nunes et al. 2021) and a line of previous studies (D'Adamo and Lupi 2021).

However, this general proposition has a number of nuances and shades deserving both high interest and deeper discussions. Namely, it appears, based on a few published indices and this newly performed formal survey, that it is possible to identify determinants for CSR supports (D'Adamo and Lupi 2021). Boldly, the age belongs to them and gender does not belong to them, while the role of the origin is not yet fully determined. Hence, it can be suggested that the increasing age and experience, perhaps as well the impact of one's own growing family and developing career, positively influence the CSR support inclination (D'Adamo and Lupi 2021). Naturally, age is linked to the earning capacity and the enhanced awareness and with a touch of exaggeration it can be proposed that older people in their productive age have more money and CSR awareness, and so they are open to pay a CSR bonus. Less obvious is the insignificance of gender and this issue should be referred to sociology and gender studies. However, for our purposes, we can satisfy ourselves that there is no gender discrimination or diversification regarding the willingness to pay a CSR bonus.

Even more interesting and, despite the prevailing stakeholder theory, the firm establishment of Carroll's pyramid and the proclaimed high digital literacy and general CSR awareness by Generation Z, there is a more than de minimis group between them totally rejecting the advanced CSR perception and stakeholder model. This group looks prima facie as an assembly of heretics who are greedy and do not care about sustainability and challenge Carroll's pyramid. However, the combination of a formal questionnaire survey and informal survey suggests that these respondents from other than classic developed countries (USA and original 12 EU members) have legitimate reasons for not paying a CSR bonus. It can be argued that it is not so much about the pro- and against-CSR but rather about belief or non-belief in CSR statements, presentations and reporting by businesses. It appears that respondents from other than classic developed countries have an issue with the institutionalized CSR, because they are concerned regarding the asymmetry of information, lack of trust, etc. Boldly, they do not believe in CSR as stated by businesses and do not want to pay money for that. However, it would be extremely incorrect to jump to the conventional conclusions about their lack of interest in CSR. The lack or insufficiency of resources can play a role as well. However, due to the just weak impact of the origin, it could be proposed that it does not contribute that much to the "unconscious" consumption. Similarly, COVID-19 is one of several factors, but definitely not the leading one—at least according to the anonymous interviews of these 18 CSR bonus-rejecting respondents.

Indeed, the finding about the lack of the impact of gender, the not as yet determined impact or lack of impact of origin, and impact of age along with the absence of any CSR bonus rejecters in the category of private students from classic developed countries along with the asymmetry of information and lack of trust leads to a message. In short, a message taking the form of a frustrated call for more transparency and veracity regarding CSR. This message is not completely new and it has been already observed that businesses take advantage of the information asymmetry (Kingston 2021) and undergo a constant

temptation to adhere only formally to institutionalized rules without putting into practice what they prescribe (Meyer and Rowan 1977; Roman et al. 2019). The conscious consumption can be done via an official institutionalized CSR channel or otherwise, e.g., by merely purchasing the cheapest product and paying with the saved money directly for a sustainability project. Businesses need to humbly recognize that and admit that a plain statement about a commitment to the Carroll's pyramid temple—CSR—does not impress *per se* all members of Generation Z. These new customers master IS/IT and look for more information in the digital setting and draw their own conclusions. Indeed, they do the content analysis by themselves (Vourvachis and Woodward 2015) and do not hesitate to double-check its application, then they gladly post their opinions online. Manifestly, those from classic developed countries believe more in the current operation of the rule of law, including the legitimacy and transparency, than others. They are aware about CSR and believe their businesses, while others are more doubtful and decline the institutionalized CSR and instead rather take the "do-it-yourself-approach".

Therefore, the conscious and unconscious consumption dichotomy in the format of CSR believers (CSR bonus payers) v. CSR heretics (not CSR bonus payers) is a demonstration of superficiality and of a total misunderstanding. Members of Generation Z want conscious consumption and are ready to go for it and contribute to it. A larger part of them, typically older members and/or those of certain origins, such as the classic developed countries, believe in CSR, trust businesses declaring a CSR commitment and pay a CSR bonus. A smaller section of members of Generation Z, typically younger members and/or those of a certain origin, such as from post-Soviet countries (Azerbaijan, Belarus, Kazakhstan, Kirghizstan, Moldova, Tajikistan, Ukraine and Uzbekistan) do not trust businesses declaring a CSR commitment and do not want to pay any CSR bonus. That is that. However, at least some of them are concerned regarding sustainability and CSR and are ready to contribute to it, but this in an individual manner (by themselves) and not in an institutionalized manner (by state and/or businesses with their CSR). This fits perfectly into the theory of the (conscious) consumption with relinquishment—buy the cheapest and invest the saved money and efforts directly in sustainability (Kingston 2021). Consequently, the rejection of a CSR bonus does not necessarily mean unconscious consumption. At the same time, the rejection of CSR is a great warning signal about the current social, business and legal settings. Indeed, it is a potential indicator about deficiencies regarding the CSR awareness, and perhaps even doubts regarding the rule of law. Therefore, the alleged CSR heretics rejecting a CSR bonus can actually be rather humble prophets facing Pharisees blindly supporting each and every business pretending a CSR commitment and asking for a stronger state involvement. Friedrich Hayek, and he is not the only one, might be instead pleased with these "heretics". Members of Generation Z get information from the media and want integrity and consistency and they are unlikely to be impressed by businesses taking a utilitarian market-based approach to CSR.

Well, we are far from making the last judgment and definitely these propositions and ideas need to undergo a thorough scrutiny by future studies entailing many more Generation Z members and checking not only what they declare to do or not to do, but as well checking what they really do and do not do and for what reasons. After all, CSR relies on the multi-stakeholder model and members of Generation Z, regardless of their origin, are stakeholders to be taken very seriously. At least forward thinking states and businesses should realize that and stop building their CSR—Carroll's pyramids on foundations of sand. COVID-19 is not an obstacle to do so, on the contrary, it is a great opportunity to change heart (D'Adamo and Lupi 2021) and to have a fresh start.

## 6. Conclusions

The conscious consumption as the individual demonstration of the commitment to the sustainability is a positive feature of the 21st century global society. States attempt to support that by setting proper legal setting and by launching pro-sustainability and pro-CSR policies and legal frameworks. Businesses attempt to match it with their CSR.

Indeed, businesses reflect the demand for sustainability by developing their economic, legal, ethical and philanthropic responsibilities and answer requirements, expectations and desires of all stakeholders, especially those with a growing purchase power—members of Generation Z. However, this does not imply *per se* that each business declaring CSR can charge more for its products and customers will gladly pay for it a CSR bonus. Further, the COVID-19 era represents a great testing opportunity, because a crisis of such a dimension magnifies already launched trends.

The assessment of a formal questionnaire, replied to by 228 private students, members of Generation Z, at a university in Prague in Spring 2021 revealed that 18 of them totally reject the CSR bonus. Contingency tables show that these 18 are recruited rather from younger respondents and from certain regions, i.e., older members of Generation Z and members of Generation Z from classic developed countries are more likely to pay a CSR bonus than other members of Generation Z. Interestingly, gender does not have any impact and so males and females from Generation Z have basically the same inclination to pay or not to pay a CSR bonus.

The assessment of additional anonymous informal interviews of these 18 provided highly relevant, and to a certain extent surprising, complementary information about several reasons for the rejection. The holistic Meta-analysis leads to an unexpected proposition: prima facie sustainability heretics declining to pay any CSR bonus can be conscious consumers and very responsible and progressive supporters of sustainability and CSR. These prima facie black sheep heretics basically reject a CSR bonus, but not sustainability and CSR as such. Indeed, they are responsible consciousness consumers in an individual and informal manner. They merely do not believe in the institutionalized conventional CSR tainted by the asymmetry of information and lack of mutual trust. Indeed, they are responsible and progressive supporters of sustainability and CSR and their CSR bonus payment rejection is a cry in the desert for more transparency, trust and rule of law. The black-and-white vision and pragmatic utilitarianism are definitely not the mantra of the curious and data-thirsty Generation Z which can be more value and principle oriented that it might seem—perhaps the deontological and integrated social contracts theory era is just at the gate.

Since the performed case study brings rather surprising, perhaps even controversial propositions and, at the same time, entails 228 respondents, Generation Z private tertiary students in Prague, from over 30 jurisdictions, it is definitely not conclusive and has inherent limitations which need to be mitigated by future studies. Firstly, a larger pool of this type of respondents needs to be included. Secondly, a different pool of respondents-consumers needs to be assessed. Thirdly, efforts need to be made to ask for and obtain the information about their reasons for supporting or not supporting sustainability and/or paying a CSR bonus. Fourthly, real life field observations have to be done to double-check the compliance of statements and actions. Fifthly, the longitudinality needs to be addressed by doing these types of studies over the course of several years. Sixthly, the structure of the survey should be expanded and additional questions included in order to employ a more advanced statistical modelling. These measures should verify and/or correct the mentioned propositions, boost their academic robustness and, most importantly, provide good advice to businesses interested in conscious consumption and conscious consumers, regardless whether they are for formal and institutionalized CSR or for an informal and individual CSR approach. Nevertheless, despite that, already the currently presented case study and indicative propositions are valuable both on theoretical and practical levels. They contribute to the fresh current academic literature stream about management and marketing and call for a reconsideration of the rank of priorities of variables in this context. Further, they provide practical advice for both management and marketing regarding e.g., the CSR bonus.

After all, the CSR relies on the multi-stakeholder model and all customers, including members of Generation Z, regardless of their origin, are stakeholders to be taken very seriously by businesses wanting to be successful in the long term. For such busi-

nesses, COVID-19 is a great opportunity for a more effective, efficient, legitimate and e-communicated CSR.

**Author Contributions:** Conceptualization, R.M.P. and M.H.; methodology, M.H.; software, M.H.; validation, M.H.; formal analysis, M.H. and R.M.P.; investigation, M.H. and R.M.P.; resources, M.H. and R.M.P.; data curation, M.H.; writing—original draft preparation, M.H. and R.M.P.; writing—review and editing, M.H. and R.M.P.; visualization, M.H.; supervision, M.H.; project administration, R.M.P.; funding acquisition, M.H. and R.M.P. All authors have read and agreed to the published version of the manuscript.

**Funding:** This research and resulting paper are the outcome of Metropolitan University Prague research project No. 87-02 "International Business, Financial Management and Tourism" (2021) based on a grant from the Institutional Fund for the Long-term Strategic Development of Research Organi-zations.

**Informed Consent Statement:** Informed consent was obtained from all subjects involved in the study.

**Acknowledgments:** The authors are grateful for the ongoing institutional support arranged by the Centre for Re-search Support at the Metropolitan University Prague, especially Tereza Vogeltanzová and Ing. Hana Raková, and highly relevant useful comments and suggestions provided during the peer-review.

**Conflicts of Interest:** The authors declare no conflict of interest.

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
