# Peer review of "CSR Unconscious Consumption by Generation Z in the COVID-19 Era—Responsible Heretics Not Paying CSR Bonus?"

_jrfm, doi:10.3390/jrfm14080390_

Round 1

Reviewer 1 Report

Dear authors,

I had the opportunity to read the paper "CSR Unconscious Consumption by Generation Z in the COVID-19 Era- Responsible Heretics not Paying CSR Bonus?" and I found it interesting.

However, there are some suggestions that you should follow. In particular:

The abstract should make clear the purpose of the paper.

Introduction: Too long. On the other hand, it is not clear the objective of the paper or the contribution of the work to the current literature. At the end of the introduction the structure of the study is not indicated.

Theoretical Background: There is no mention of bibliography related to the variables that are used in section 3 Materials and Methods: gender, age, origin. There should be more relationship between the two sections.

Materials and Methods: How the Delphi method was implemented should be better explained.

Results: the results of the Delphi method should be better explained.

In conclusions: Limitations and future research are lacking.

In references: There is some references with the year in parentheses.

Author Response

FIRST REVIEWER

Dear authors,

I had the opportunity to read the paper "CSR Unconscious Consumption by Generation Z in the COVID-19 Era- Responsible Heretics not Paying CSR Bonus?" and I found it interesting.

However, there are some suggestions that you should follow. In particular:

The abstract should make clear the purpose of the paper.

You are right, we have updated the abstract accordingly.

Introduction: Too long. On the other hand, it is not clear the objective of the paper or the contribution of the work to the current literature. At the end of the introduction the structure of the study is not indicated.

Sure, the information about aim and structure was added.

Theoretical Background: There is no mention of bibliography related to the variables that are used in section 3 Materials and Methods: gender, age, origin. There should be more relationship between the two sections.

You are absolutely right. We have added this important information along with a number of references to our Theoretical Background.

Materials and Methods: How the Delphi method was implemented should be better explained.

Yes, we have updated our paper accordingly.

Results: the results of the Delphi method should be better explained.

Yes, we have updated our paper accordingly.

In conclusions: Limitations and future research are lacking.

It is included in the last paragraph, i.e. lines 700+.

In references: There is some references with the year in parentheses.

Thanks for catching this and letting us know, we have corrected it.

Reviewer 2 Report

  1. Originality

Introduction does not clarify the objectives of the investigation neither the underlying argument. In fact, no gaps have been identified and the “what” and “why” remain unclear.

Relationship to Literature

The background is confuse and difficult to follow. It is a long text, without sections, without building an argument, without developing hypotheses (which are then presented in the results), without our being able to understand the objectives in question.

The consumption and buying behavior of generation Z is not presented and/or discussed. The stakeholder theory does not seem at all adequate to develop the argumentation that we can intuit at the end.

Marketing literature on customer behavior was ignored.

Methodology:

Convenience sampling was used based on a specific type of students from business and law: “students taking paid tertiary education”.

The question raised is too general, does not focus on a product or service, so the results are more than doubtful: “”Considering the current situation 396 and global society challenges, would you please indicate how much extra in % you are open to pay for an identical product/service of a business which goes strongly for sustainability and CSR as  opposed to a neutral business, i.e. doing nothing for or against sustainability and CSR (nothing for  or against the environment, neutrally treating employees, neither helping nor damaging society, etc.). Hence, there is no difference between the compared outcomes, they have the same quality and they differ only regarding their sources and you might be open to pay e.g. 20% extra in the case of a business going strongly for CSR … or maybe you do not want to pay anything extra. It does not matter, just please indicate below your honest opinion.”. How to interpret the answers? The question is not clear at all; The question does not allow to identify the kind of product or service that customer have in mind; Customers are not invited to think about a brand, a product, a category of products or services… nothing: only and hypothetical desire to pay an extra for… we do not know what. What is a bonus? With what objectives? A cause? A different product/service performance? Environmental protection?

The statistical modelling is basic and the answers provided are limited. What we have are averages comparisons, using chi square.

  1. Results and contributions

Results section presents the test of hypotheses that were not raised or theoretically supported.

No contributions were presented. In fact, it is hard to identify any relevant contribution.

  1. Quality of Communication:

The paper is well written, but the jargon used is very poor or even not correct.

Author Response

SECOND REVIEWER

Introduction does not clarify the objectives of the investigation neither the underlying argument. In fact, no gaps have been identified and the “what” and “why” remain unclear.

This is a good point and we have significantly updated our introduction accordingly. Hopefully, the new version meets your expectations.

Relationship to Literature

The background is confuse and difficult to follow. It is a long text, without sections, without building an argument, without developing hypotheses (which are then presented in the results), without our being able to understand the objectives in question.

The consumption and buying behavior of generation Z is not presented and/or discussed. The stakeholder theory does not seem at all adequate to develop the argumentation that we can intuit at the end.

Marketing literature on customer behavior was ignored.

Your objections are relevant and constructive. Therefore, we have significantly updated our literature section. In particular, we have added information tying this section to other parts of the paper as well as information about Generation Z and its behavior. We have included various recent propositions and (semi)conclusions implied by the marketing literature. Ultimately, this leads to the expansion of this section of our paper and to an increase in the number of references by 15 high quality papers. Undoubtedly, the academic depth was improved and we are grateful to you for your insight and advice, which made us to do so.

Methodology:

Convenience sampling was used based on a specific type of students from business and law: “students taking paid tertiary education”.

The question raised is too general, does not focus on a product or service, so the results are more than doubtful: “”Considering the current situation 396 and global society challenges, would you please indicate how much extra in % you are open to pay for an identical product/service of a business which goes strongly for sustainability and CSR as  opposed to a neutral business, i.e. doing nothing for or against sustainability and CSR (nothing for  or against the environment, neutrally treating employees, neither helping nor damaging society, etc.). Hence, there is no difference between the compared outcomes, they have the same quality and they differ only regarding their sources and you might be open to pay e.g. 20% extra in the case of a business going strongly for CSR … or maybe you do not want to pay anything extra. It does not matter, just please indicate below your honest opinion.”. How to interpret the answers? The question is not clear at all; The question does not allow to identify the kind of product or service that customer have in mind; Customers are not invited to think about a brand, a product, a category of products or services… nothing: only and hypothetical desire to pay an extra for… we do not know what. What is a bonus? With what objectives? A cause? A different product/service performance? Environmental protection?

You are right, these questions are highly legitimate and relevant. In our original paper draft we skipped the information explaining further the manner in re how the survey was performed and other important aspects. Based on your objections, we have revisited this part of our paper and have added the missing information.

The statistical modelling is basic and the answers provided are limited. What we have are averages comparisons, using chi square.

Yes, we have not engaged in a complex statistical modeling and our research and resulting paper are rather indicative than conclusive. We have updated our draft and underlined that in our Discussion and Conclusions with limitations.

Further, your objections induce us to think about a forthcoming survey with more questions and allowing a more advanced statistical modelling.

Results and contributions

Results section presents the test of hypotheses that were not raised or theoretically supported.

Since our introduction and literature sections have been dramatically expanded and information related to the tested factors have been added, we think that this objection is now taken care of.

No contributions were presented. In fact, it is hard to identify any relevant contribution.

You are right, in our prior draft this was not sufficiently obvious, hence we have added an explicit part about it at very end of our conclusions.

Quality of Communication:

The paper is well written, but the jargon used is very poor or even not correct.

Thanks, we have taken care of that.

Reviewer 3 Report

INTRODUCTION

  • A large part of the Introduction refers to historical facts, from Ancient Egypt to the Hanseatic League, which have allegedly contributed to spread sustainability. However, it is not clear how this happened. The Authors should explain better, or eliminate that part.
  • Lines 52-54: The 1948 UDHR certainly considers and states the social pillar of sustainability. On the contrary, the environmental and economic pillars are not covered by the UDHR. The concept should be expressed in a better way. The three pillars can be linked to the Brundtland Report, mentioned in Line 73.
  • Lines 68-80: There is gap between the Brundtland Report (1987) and the 2030 Agenda (2015). For example, at least the Global Compact and the Millennium Development Goals (both adopted in 2000) should be mentioned.
  • Line 106: You should explain why CSR and codes of ethics are answers to the current Covid-19 pandemic.

SEC. 2 – THEORETICAL BACKGROUND

  • Line 227: When I read that «a good CSR has “5Ps”», I expected to find 5 characteristics all starting with a P. However, the elements listed from (i) to (v) don’t comply with this. So, does “5Ps” stand for “5 pillars”?

SEC. 3 – MATERIALS AND METHODS

  • Lines 375-377: I don’t think it surprising or even shocking that young people who are still attending university, and probably don’t have a stable job, have declared to be reluctant to pay a CSR bonus. Young students may have ideals, but they don’t have so much money to spend for those ideals.
  • Lines 392-393: I would have asked the students to specify their (o their family’s) income level too.
  • Line 437: “… such a CSR bonus” (add “bonus” to the sentence).
  • Line 444: A brief description of the Delphi method, with its strengthens and weaknesses, would be appreciated.

SEC. 4 – RESULTS

  • Your paper refers to “Generation Z” (i.e. persons born between 1995 and 2010). However, Table 1 shows that you interviewed 39 students aged 26-50. They don’t belong “Z-gen”.

SEC. 5 – DISCUSSION

  • Lines 523-525: If members of Z-gen who are paying to attend a private university in Prague are also “financially strong and solvent consumers”, you should write this statement before. This could be a characteristic of your sample.

PUNCTUATION AND TYPOS:

  • Lines 19, 455, 644, 646: I think it should be “prima facie” (instead of “facia”).
  • Line 21: There are two full marks at the end of the line.
  • Lines 23-24: Use semicolons to separate the keywords.
  • Line 182: “similar to” (separate the two words).
  • Line 240: two commas have been typed one after the other.
  • Line 303: 1870’s
  • Line 315: …to make a set… (delete “a” before “make”).

Author Response

THIRD REVIEWER

INTRODUCTION

A large part of the Introduction refers to historical facts, from Ancient Egypt to the Hanseatic League, which have allegedly contributed to spread sustainability. However, it is not clear how this happened. The Authors should explain better, or eliminate that part.

You are absolutely right and we have provided a better explanation about the synergetic interrelation of the order – consumption – liability which leads to the concept of sustainability.

    Lines 52-54: The 1948 UDHR certainly considers and states the social pillar of sustainability. On the contrary, the environmental and economic pillars are not covered by the UDHR. The concept should be expressed in a better way. The three pillars can be linked to the Brundtland Report, mentioned in Line 73.

Yes, our original presentation might sound misleading in this aspects and so we have rephrased it.

    Lines 68-80: There is gap between the Brundtland Report (1987) and the 2030 Agenda (2015). For example, at least the Global Compact and the Millennium Development Goals (both adopted in 2000) should be mentioned.

OK, We have expanded the information accordingly, but due to the size limit of the paper, we have not engaged in any deeper development in this respect.

    Line 106: You should explain why CSR and codes of ethics are answers to the current Covid-19 pandemic.

Sure, we have gladly explained it.

SEC. 2 – THEORETICAL BACKGROUND

    Line 227: When I read that «a good CSR has “5Ps”», I expected to find 5 characteristics all starting with a P. However, the elements listed from (i) to (v) don’t comply with this. So, does “5Ps” stand for “5 pillars”?

Point taken, we have corrected it..

SEC. 3 – MATERIALS AND METHODS

    Lines 375-377: I don’t think it surprising or even shocking that young people who are still attending university, and probably don’t have a stable job, have declared to be reluctant to pay a CSR bonus. Young students may have ideals, but they don’t have so much money to spend for those ideals.

OK, we have specified that it is about a solvent segment and have removed the “shocking”.

    Lines 392-393: I would have asked the students to specify their (o their family’s) income level too.

 Yes, it would be interesting to know, but it was not feasible, i.e. OK, the majority of students have declined to provide such an information.

   Line 437: “… such a CSR bonus” (add “bonus” to the sentence).

Yes, done, thanks.

    Line 444: A brief description of the Delphi method, with its strengthens and weaknesses, would be appreciated.

Sure, we have added this information.

SEC. 4 – RESULTS

    Your paper refers to “Generation Z” (i.e. persons born between 1995 and 2010). However, Table 1 shows that you interviewed 39 students aged 26-50. They don’t belong “Z-gen”.

Thanks for catching this. Firstly, we have corrected our typo (26+,i.e. 26-28 and not 26-50), secondly we have provided a short comment explaining the inclusion of this border group which might be perceived rather as millennials.

SEC. 5 – DISCUSSION

    Lines 523-525: If members of Z-gen who are paying to attend a private university in Prague are also “financially strong and solvent consumers”, you should write this statement before. This could be a characteristic of your sample.

Thanks, done.

PUNCTUATION AND TYPOS:

    Lines 19, 455, 644, 646: I think it should be “prima facie” (instead of “facia”).

    Line 21: There are two full marks at the end of the line.

    Lines 23-24: Use semicolons to separate the keywords.

    Line 182: “similar to” (separate the two words).

    Line 240: two commas have been typed one after the other.

    Line 303: 1870’s

    Line 315: …to make a set… (delete “a” before “make”).

Thanks, your concentration and attentiveness are perfect. We have corrected our paper accordingly. The only exception is “1870 ´”, there we truly want to keep “1930 ´”, because it is about the Great Depression.

Round 2

Reviewer 1 Report

Dear authors: 

My recommendation is to accept your paper in the present form. 

King regards,

Reviewer 3 Report

The Authors accepted my suggestions. Thanks.

Please, correct two typos: 

  • line 82: there is a double open bracket;
  • line 259: too many dots. Perhaps you intended to insert a source.